# First-Principles Study of the Magnetic and Electronic Structure of NdB_4_

**DOI:** 10.3390/ma16072627

**Published:** 2023-03-26

**Authors:** Pengyan Tao, Jiangjiang Ma, Shujing Li, Xiaohong Shao, Baotian Wang

**Affiliations:** 1College of Mathematics and Physics, Beijing University of Chemical Technology, Beijing 100029, China; 2Institute of High Energy Physics, Chinese Academy of Sciences (CAS), Beijing 100049, China; 3Spallation Neutron Source Science Center (SNSSC), Dongguan 523803, China; 4Collaborative Innovation Center of Extreme Optics, Shanxi University, Taiyuan 030006, China

**Keywords:** non-collinear magnetism, first-principles calculation, magnetic ground state, electronic properties

## Abstract

Due to their magnetic and physical properties, rare earth magnetic borides have been applied to a variety of critical technologies. In particular, rare earth tetraborides are more abundant as frustrated antiferromagnets. Here, the atomic structures, magnetic structures, and electronic structures of NdB_4_ have been studied by first-principle calculations. The ground state magnetic structure of NdB_4_ is determined. Moreover, the small energy difference between different magnetic structures means that there may be more than one magnetic structure that coexist. One can glean from the electronic structure of the magnetic ground state that the d orbital of Nd is strongly hybridized with the *p* orbital of B, and the *f* electron of Nd is highly localized. The computational results reveal the complexity of the magnetic structure and provide a theoretical basis for studying the magnetic ground state of NdB_4_.

## 1. Introduction

Boron-rich borides encompass a fascinating range of materials showcasing distinct structural and physical properties. Significantly analogous to carbon, boron exhibits similar clustered and stratified structural features, as observed in fullerenes and graphite intercalation compounds [1]. The combination of boron with metal atoms exhibits significant and extensively investigated properties. A prime illustration is MgB_2_, considered as the foremost phonon-coupled superconductor [2]. Another class of hexaborides, MB_6_, formed by divalent metals, is worthy of attention. This category comprises divalent metals such as Ca, Sr, and Ba, which are considered to be small-gap semiconductors [3,4,5,6]. Additionally, the rectangular boron units in CrB_4_ and MnB_4_ provide exceptional mechanical properties, such as high stiffness and ideal strength [7].

The boron and rare earth (RE) compounds are a fascinating class of materials with a wide range of physical properties and potential applications, such as thermal electron emission, chemical stability, hardness, high melting point, magnetic properties, low work function, narrow-band semiconductor behavior, and superconductivity [4,5,8,9]. For instance, RE borides, such as SmB_6_, PmB_6_, NdB_6_, and EuB_6_, are of particular interest due to their topologically nontrivial behavior and potential for applications in quantum computing and electronics [10]. Doping of RE borides is also a hot topic in recent years, with researchers investigating the effects of different dopants on their electronic and magnetic properties [11,12]. Studies have also shown that certain RE borides, such as $R\bar{3}m$-YB_6_, have potential as ultrahard metals with high Vickers hardness [13]. In addition, the development of high entropy RE hexaborides/tetraborides (HE REB_6_/HE REB_4_) composite ceramics is a promising area of research for high-performance electromagnetic wave absorbing materials [14]. REB_4_ compounds are hard and refractory materials, making them ideal for use as coatings for cutting tools, turbine blades, and other high-wear components. They also exhibit good thermal conductivity and resistance to oxidation, making them useful in high-temperature applications [15].

In metallic compounds of rare earth (RE), the most prominent interaction that is responsible for the alignment of the magnetic moments of RE ions is the indirect Ruderman–Kittel–Kasuya–Yosida (RKKY) [16,17,18] exchange. This coupling demonstrates an oscillatory and long-range character, which frequently opposes other interactions, such as the crystal electric field (CEF) magnetoelastic coupling or quadrupolar interactions, causing competing interionic interactions to arise [19]. In addition, adding a specific crystal structure will form a complex frustrated magnetic system, especially a structure formed by the RE tetraborides (REB_4_), all interactions between adjacent spins cannot be satisfied at the same time, suppressing the magnetic order, resulting in ground state degradation, i.e., several different magnetic states with the same energy are formed.

In particular, REB_4_, as frustrated systems, have attracted considerable attention due to their complex magnetic phases. In earlier results, Yin et al. analyzed the electronic structure of some tetraboride systems by using the local density approximation (LDA) [20]+ U method [21]. The tetragonal structure of the REB_4_ family (excluding Eu and Pm among the lanthanides), which is a complex configuration consisting of squares and equilateral triangles on the ab-plane, exhibits a striking resemblance to the Shastry-Sutherland lattice (SSL) [22] (see Figure 1a). Similar to REB_4_, neodymium tetraboride (NdB_4_) also has unusual magnetic properties. So far, successive phase transitions of NdB_4_ have been successfully observed at *T*_0_ = 17.2 K, *T*_N1_ = 7.0 K, and *T*_N2_ = 4.8 K [23]. The magnetic structure of the phase Ⅱ ($T_{N1}<T<T_{0}$) was identified as a linear combination of antiferromagnetic (AFM) structures, “all-in/all-out”-type and “vortex”-type structures [24]. It is worth noting that the magnetic transition temperature of REB_4_ does not adhere to the de Gennes scaling factor [25]. In other words, in the isomorphic series where RE atoms are the only magnetic constituents, the magnetic transition temperature is directly proportional to ${\left(g_{J}-1\right)}^{2}J\left(J+1\right)$, where *J* represents Hund’s rule total angular momentum index and *g_J_* represents the corresponding Landég factor. This indicates that the magnetic properties of REB_4_ are very complex. Several magnetic moment orientations have been observed below the Neel temperature in various REB_4_, accompanied by multiple phase transitions. Such behavior implies that they are responsive to the unique intricacies of their electronic structures. As suggested by several research studies [24,26,27], it has been suggested that the magnetic moment of Nd^3+^ ions is ordered independently along the $\vec{c}$ and $\vec{a}$,$\vec{b}$ components at varying temperatures. The magnetic structure factors ${\vec{q}}_{ab}=\left(0.2,0,0.1\right)$ and ${\vec{q}}_{c}=\left(0,0,0.5\right)$ and an incommensurate modulation of the magnetic moment along the *c*-axis in phase IV ($T<T_{N2}$) were measured by polarized neutron diffraction [26].

Previous studies have shown the unusual magnetic properties of Nd tetraboride, while few studies have focused on the effect of electronic structure on the magnetic properties. Therefore, this work provides theoretical support for this material by investigating the magnetic and electronic structure of this typical magnetic boride, mainly through first principles, predicting the magnetic ground state and investigation the contribution of *f*-electrons to the magnetic properties.

## 2. Computational Methods

We have conducted calculations for all feasible magnetic structures based on the density functional theory (DFT) within the generalized gradient approximation (GGA) [28], utilizing the Perdew–Burke–Ernzerhof (PBE) [29] exchange-correlation functional, as implemented in the Vienna Ab-initio Simulation Package (VASP) [30,31]. The ion-electron interaction is described by projected augmented wave (PAW) [32] potentials, while considering spin-orbit coupling (SOC) and magnetic noncollinearity, as described by Hobbs et al. [33]. The plane wave set is defined with a cutoff energy of 520 eV, and Brillouin zone (BZ) integrations are performed using the Monkhorst-Pack (MP) [34] method with a regularly spaced mesh of 3×3×6. We calculate the density of state (DOS) with a k point-mesh of 7×7×12. The atomic positions and lattice vectors are relaxed until the residual forces are less than −0.01 eV/Å. We have set the energy convergence criterion to 10^−7^ eV/atom. In all cases, the structures are fully relaxed concerning volume as well as cell-internal and cell-external coordinates.

The 4*f* electrons in Nd are relatively localized, resulting in extraordinarily complex non-collinear magnetic conditions with multiple local minimum energies, which require the use of initial structural relaxation in magnetic conditions. For the above reasons, we include SOC in the second variational method of the scalar relativistic wave function [35,36,37,38]. Fourteen different initial magnetic moments are calculated and named as 1-AFM, 2-AFM, 3-AFM, 4-AFM, etc. The first four magnetic moments used in the calculations are determined from [24]. In the geometrical optimizations, the initial magnetization directions are set as small terms of the irreducible representations and are associated with the basis vectors of the Nd site in the space group *P4/mbm* (Figure 2). When the structures are fully relaxed, each magnetic vector eventually relaxes to each independent direction.

In order to account for the strong correlation among 4*f* electrons, we used GGA + U [39] method. In this study, the effective parameter U_eff_ = U-J was employed, where U and J are the Coulomb and exchange parameters. The Coulomb U is treated as a variable, while the exchange energy is set to be a constant J = 0.6 eV. Since only the difference between U and J is significant, for convenience, the effective parameter U_eff_ will be referred to as U in the following paper. We discuss our results in the following sections.

## 3. Results and Discussion

### 3.1. Crystal and Magnetic Structures

Figure 1 displays the tetragonal crystal structure of NdB_4_, which consists of layers of Nd^3+^ ions arranged in a square lattice geometry and weakly coupled to each other. The crystal structure belongs to space group $D_{4h}^{5}-P4/mbm$. Table 1 lists the crystallographic sites of atoms in NdB_4_. The symmetry of the Nd site plays a crucial role in determining the magnetic properties of the compound by controlling the crystal field splitting of the ion with total angular momentum $\vec{J}=\vec{L}+\vec{S}$, and consequently, the magnetic state at low temperatures. The NdB_4_ unit cell exhibits a square arrangement of the four Nd sites, tilted at an angle of approximately 15° relative to the square sublattice of B_6_ octahedra. The B atoms form an interconnected three-dimensional network. The Nd atoms are located both above and below the center of the seven-membered boron ring in the densely packed plane of boron atoms. The top view of the NdB_4_ lattice (Figure 1) displays Nd sublattices in the ab-plane arranged in a pattern of squares and triangles, which is topologically equivalent to the SSL. The layered structure of neodymium and boron atoms is clearly visible from the side view. The boron sublattice of tetraborides is composed of diborides, which are chains of boron atoms, and hexaborides, which consist of B_6_ octahedra. The strong covalent bonding in the boron sublattice imparts tetraborides with high hardness and a high melting point. To investigate the magnetic ground state of NdB_4_, we compute the energies of various magnetic configurations as illustrated in Figure 2. The calculated results of energies and magnetic moments are tabulated in Table 2. It is clear that NdB_4_ favors the 6-AFM configuration. Our calculated and previously reported experimental lattice constants of NdB_4_ are presented in Table 2. As shown, our calculated values are slightly smaller than those reported from experiments. The differences in different magnetic states are very small.

Comparing the energies of conventional ferromagnetic (FM) and antiferromagnetic (AFM) configurations of the lattice allows for the determination of the ground state magnetic properties of the system. The different structural behaviors of the fourteen kinds of magnetic structures provide clues to pursuing the origin of the AFM order in NdB_4_. Based on our calculated data (refer to Table 2), it is evident that the compound’s ground state remains stable in the AFM phase. Additionally, the calculated total magnetic moment is the aggregate of the spin and orbital moments at the Nd atoms. Every magnetic moment in all the cases is around 1.50 μB suggests that the magnetic moment of Nd atoms is not strongly affected by different magnetic structures. In addition, the energy difference between the FM structure and the AFM structure is about 15 meV/f.u., and the energy difference between the AFM structure in the $\vec{a}$ direction and the AFM structure in the $\vec{b}$ or $\vec{c}$ direction is approximately 1 meV/f.u. The small energy difference between the FM and AFM structures indicates that the ground state magnetic structure of NdB_4_ may be more complicated than either of these two structures. Therefore, further experiments, such as neutron diffraction, are necessary to confirm the magnetic ground state. Of course, there may coexist different magnetic states. Our present results reasonably explain the doubt that there are multiple magnetic structural components in the experiment [27].

### 3.2. Electronic Structures

In this study, both SOC and non-spin-orbit coupling (non-SOC) calculations are conducted. The magnetic moments used in the calculations are the 6-AFM and the 14-FM. The GGA + U and GGA + U + SOC of FM and AFM phases reveal a correlation between the total energy (per formula element at the corresponding optimal geometry) and U, as demonstrated in Figure 3. The band gap can effectively compare the experimental and theoretical results and adjust the U value due to its obvious characteristics. Therefore, without the band gap, it is difficult to find a reasonable U value for NdB_4_. At U = 0 eV, we have determined the ground state of NdB_4_ to be an antiferromagnetic metal, in agreement with the experimental findings. The total energy of FM + SOC increases significantly when U is from 4 to 7 eV. Notably, the magnetic moments do not change much when U is increased. By comparing the two methods, we found that both GGA + SOC and GGA + SOC + U have no effect on the magnetic moment of Nd, which is consistent with the view of Nd in NdNiO_2_ [43]. Therefore, the calculation method we use is reasonable and effective.

Figure 4 and Figure 5 depict the total DOS as well as the projected DOS onto the Nd and B orbitals of AFM and FM coupled NdB_4_ at selective values of U. Typically, the application of parameter U leads to the division of the occupied and unoccupied states of the relevant atom. The incorporation of U results in a downward shift of the Nd orbitals below the Fermi energy level and an upward shift of those above the Fermi energy level to the higher energy region.

From Figure 4 and Figure 5, it is known that the Hubbard U value, whether with SOC, has relatively greater influence on the electronic structure near the Fermi level of NdB_4_. Compared with the experimental results [40], the calculated lattice constant accords well with 2 eV of the Hubbard U value. In addition, compared with the calculated results of DOS with SOC in Figure 4 and Figure 5, Figure 6 shows the calculation of DOS without SOC. It is clear that the DOS of collinear calculations without SOC are the same as those with SOC. Therefore, U = 2 eV is adopted in the following studies.

For each of the NdB_4_ magnetic structures, we conducted calculations for their energy band structure and DOS; the findings are presented in Appendix A, Figure A1 and Figure A2. As demonstrated in Table 2, NdB_4_ displays a range of magnetic configurations with an energy difference of 100 meV, indicating the existence of a complex magnetic state. It is anticipated that magnetic phase transition could arise in response to external factors such as temperature or magnetic field. Figure 7 and Figure 8 exhibit the energy band and DOS of GGA + SOC and GGA + SOC + U, respectively. Upon comparison with Figure 7, we observe several additional energy bands intersecting the Fermi level in M-Γ. We also calculate the energy bands and DOS for U = 6 eV (See Appendix B, Figure B1). As is seen, there is little difference between the two electronic structures. Moreover, the localized electrons of the Nd-4*f* orbitals are dissociated from the Fermi level. According to Figure 8, the electronic orbitals near the Fermi level of the most stable magnetic state are mainly occupied by the Nd-4*f* orbitals. The contributions from Nd-5*d* and B-2*p* orbitals are neglectable. For general RE metals, the DOS of the *f* orbitals exhibiting a strong peak indicates that the *f* electrons are relatively localized, and the corresponding energy band is relatively narrow. For NdB_4_, 4*f* orbitals contribute a big peak in valence band and also a big peak in conduction band. The Fermi level lies in between these two big peaks. Thus, the 4*f* orbitals are localized and are responsible for their metallicity. For 5-AFM, 6-AFM, 8-AFM, and 9-AFM, their electronic occupations at the Fermi level are smaller than those of other magnetic states. These results are consistent with their energy results. Under low temperature conditions, these magnetic states should be more possibly observed in experiments. 

From Figure 8, it is evident that the non-collinear configuration of 6-AFM exhibits a metallic nature. Four bands intersect the Fermi level, with some being near Γ-M points, electron pockets being formed. Another is along Γ-Z, forming the hole pockets. In conduction bands above the Fermi level, there are several flat bands, which are mainly composed of localized Nd-*f* electronic states and a very small part of the non-local Nd-*d* electronic states. In valence bands, there exist similar flat bands. Between −1 and 2 eV, the *d* orbitals of Nd atoms and *p* orbitals of B atoms exhibit significant overlap and similarity, indicating their hybridization characteristic. The bands of Nd-4*f* can be recognized by their flat and weakly hybridizing nature. The separation of the majority and minority itinerant bands offers a direct measurement of the Kondo coupling between the 4*f* moment and the band states.

### 3.3. Fermi Surface

The Fermi surface, which delineates the boundary between filled and unfilled states in the reciprocal space, plays a significant role in metallic systems. In the case of NdB_4_, the Fermi surface has been computed, and the outcome is presented in Figure 9. The color indicates the electron velocity. As seen, the electron pocket centered at Γ is in shape of ellipsoid. There is no Fermi surface in the large area of the R-A-R along the edge of the BZ, consistent with the results of the band structure. At middle of the Γ-Z, two symmetric square pyramids are present with only the middle vertex partially circular. Additionally, two pointed semiellipsoidal surfaces appear symmetrically at the Z-point. Two nested surfaces around the S-point can also be seen, which may be linked to RKKY coupling. The shape of these Fermi surfaces bears resemblance to that of YB_4_ [21] and could be correlated to the magnetic behavior observed in REB_4_ compounds.

## 4. Conclusions

In summary, this study presents a comprehensive analysis of the properties of NdB_4_, shedding light on its fundamental characteristics. The atomic, electronic, and magnetic structures of the metallic tetra-boride system NdB_4_ have been calculated using the DFT scheme. The investigation of the magnetic structure reveals that the magnetic properties of NdB_4_ are strongly influenced by magnetic frustration and quadrupole interaction. Near the Fermi level, the electronic states are mainly occupied by the Nd-*f* orbitals. Furthermore, we have predicted the in-plane magnetic ground state of NdB_4_ as the 6-AFM configuration. From comparing with FM and other AFM configurations, the magnetic structures of NdB_4_ are complex and may easily change with external conditions. NdB_4_ is a promising material for potential magnetic applications. Our results can be used as a theoretical basis for further study of magnetic and electronic structures of RE borides and this provides a theoretical basis for further study of magnetic ordering and behavior at different temperatures and magnetic fields. Furthermore, the complex magnetic interactions and crystal structure of NdB_4_ make it a good candidate for theoretical and computational modeling. Such work could help to better understand the properties and behavior of the material and guide future experimental research.

## Figures and Tables

**Figure 1 materials-16-02627-f001:**
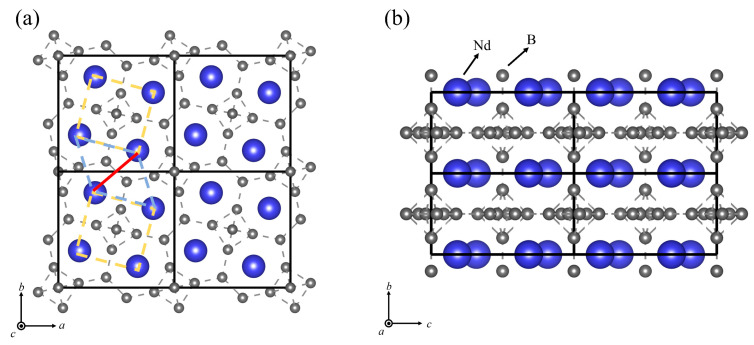
(**a**) Top view of the NdB_4_ showing the frustrated Shastry-Sutherland lattice (SSL). (**b**) Side view of the crystal structure. (Nd and B are represented by blue and grey spheres, respectively.)

**Figure 2 materials-16-02627-f002:**
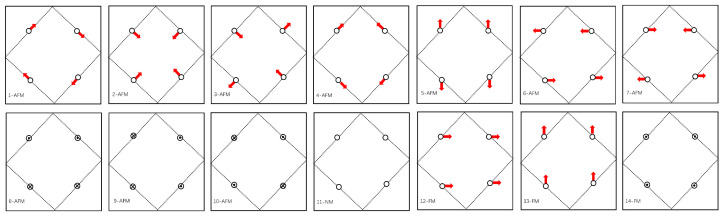
Decomposed sets in terms of the associated basis magnetic vectors of Nd.

**Figure 3 materials-16-02627-f003:**
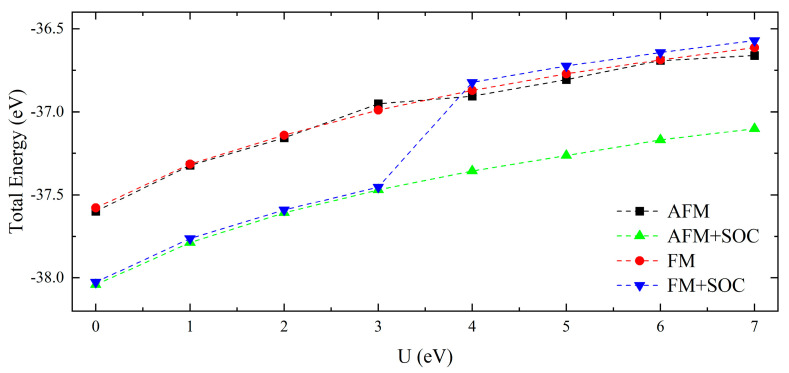
Dependences of the total energies (per formula unit) on the values of U and SOC for both AFM and FM coupled NdB_4_.

**Figure 4 materials-16-02627-f004:**
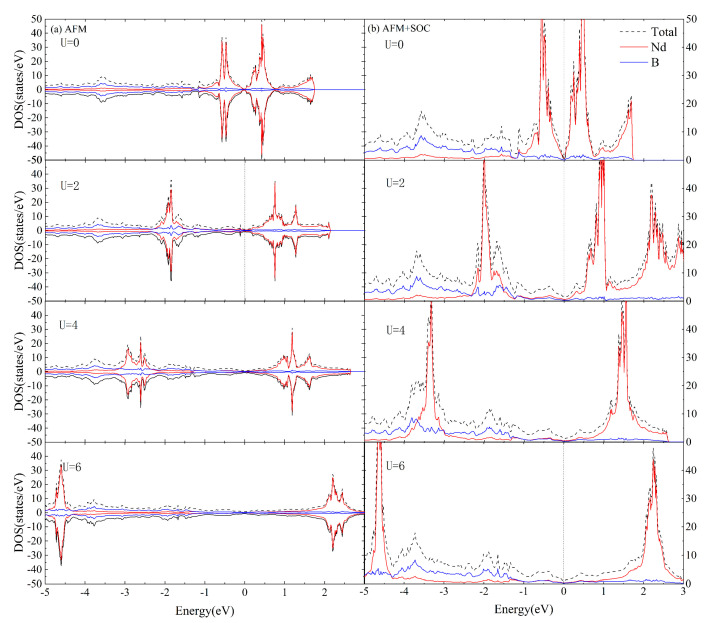
The total DOS for the magnetic phase of NdB_4_ calculated in the (**a**) AFM + U and (**b**) AFM + SOC + U formalisms with four selective values of U. The projected DOS for the Nd and B orbitals are also presented. The Fermi level is set at zero.

**Figure 5 materials-16-02627-f005:**
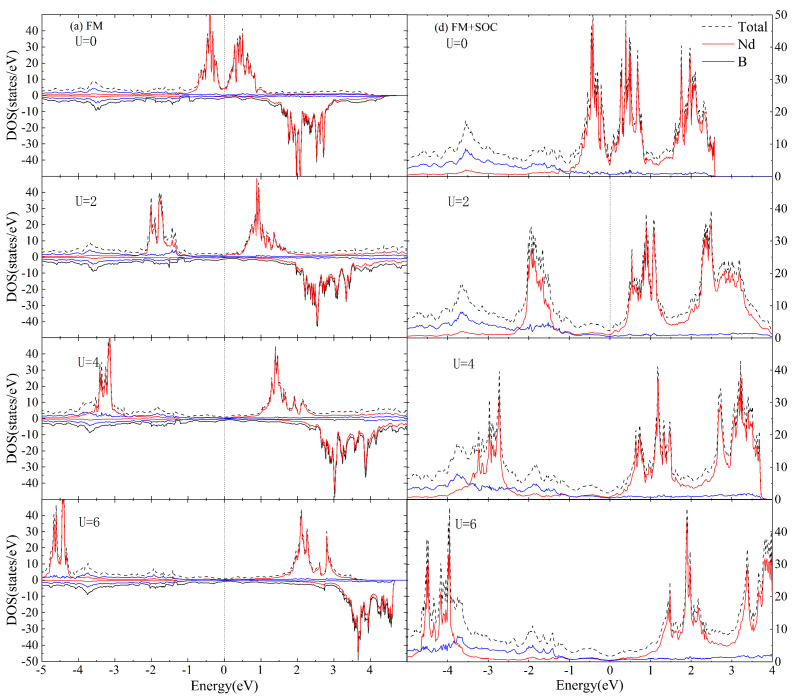
The total DOS for the magnetic phase of NdB_4_ calculated in the (**a**) FM + U and (**b**) FM + SOC + U formalisms with four selective values of U. The projected DOS for the Nd and B orbitals are also presented. The Fermi level is set at zero.

**Figure 6 materials-16-02627-f006:**
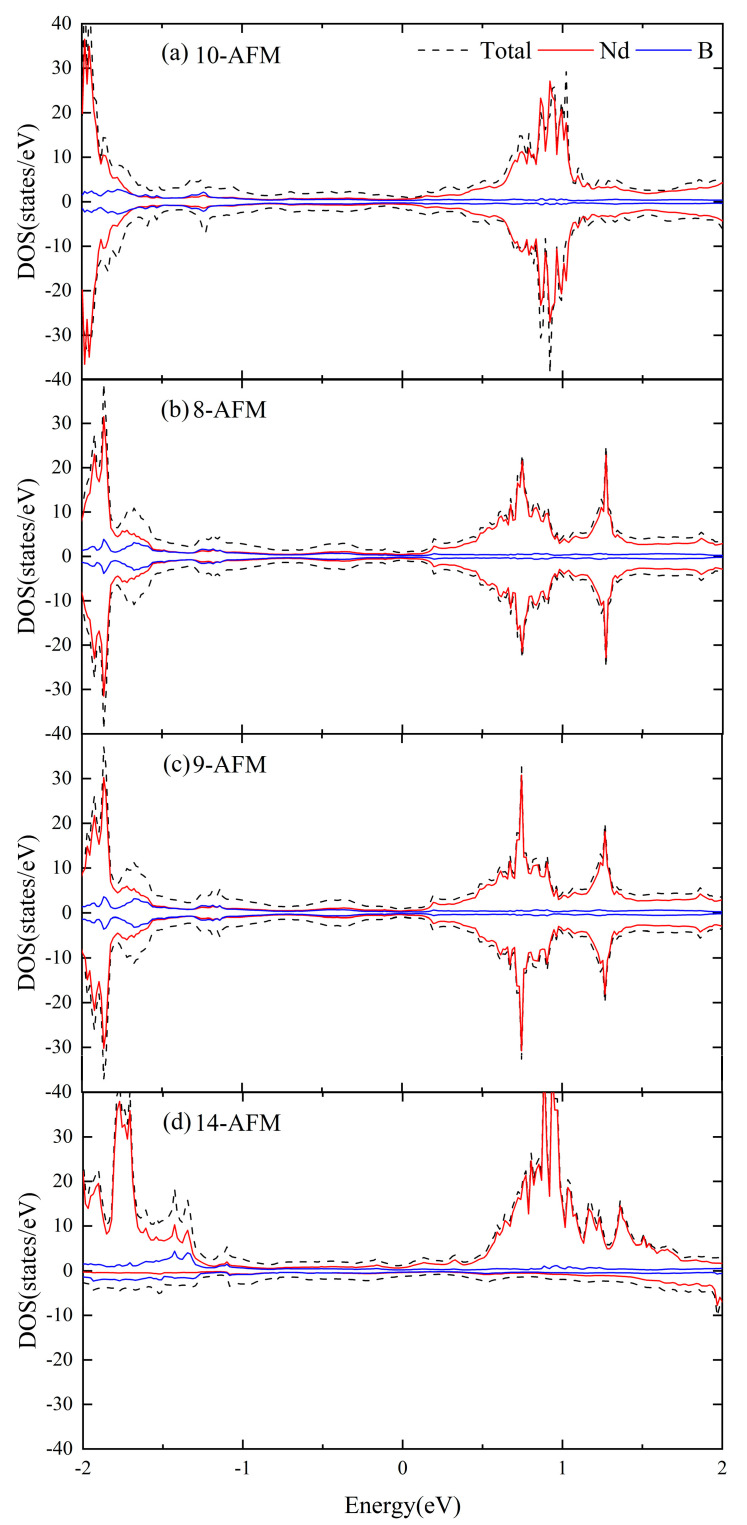
(**a**–**d**) The DOS of NdB_4_ different magnetic structures with U = 2 eV and non-SOC. The projected DOS of the Nd and B orbitals are also presented. The Fermi level is set at zero.

**Figure 7 materials-16-02627-f007:**
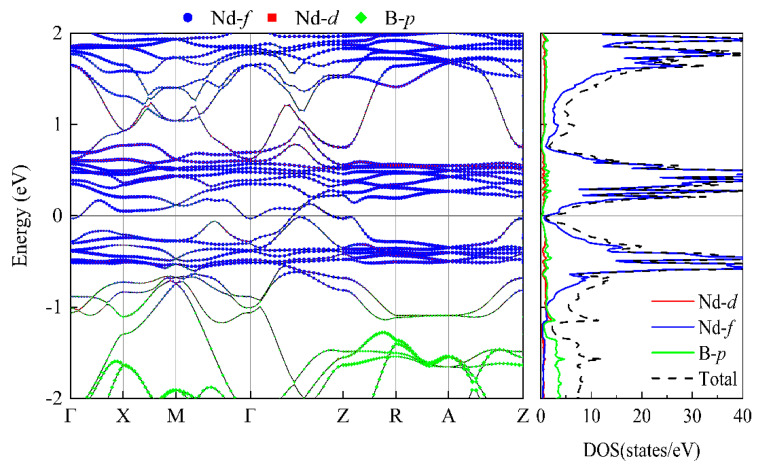
The energy bands and DOS of the 6-AFM configuration with U = 0 eV and SOC. The Fermi level is set to zero.

**Figure 8 materials-16-02627-f008:**
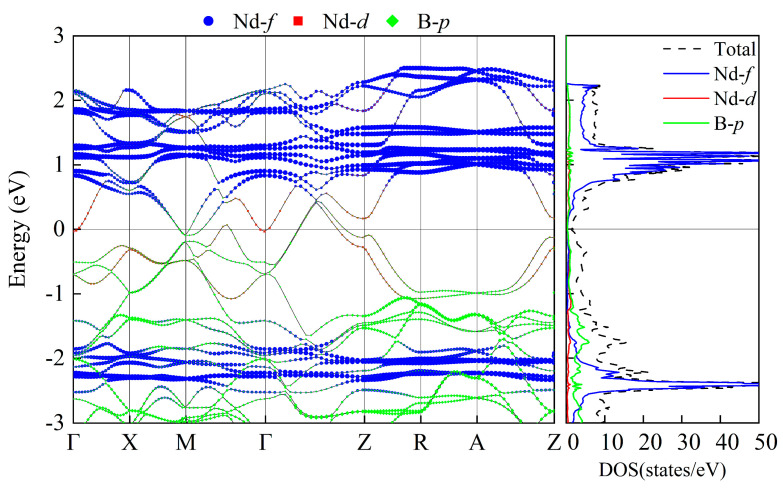
The energy bands and DOS of the 6-AFM configuration with SOC and U = 2 eV.

**Figure 9 materials-16-02627-f009:**
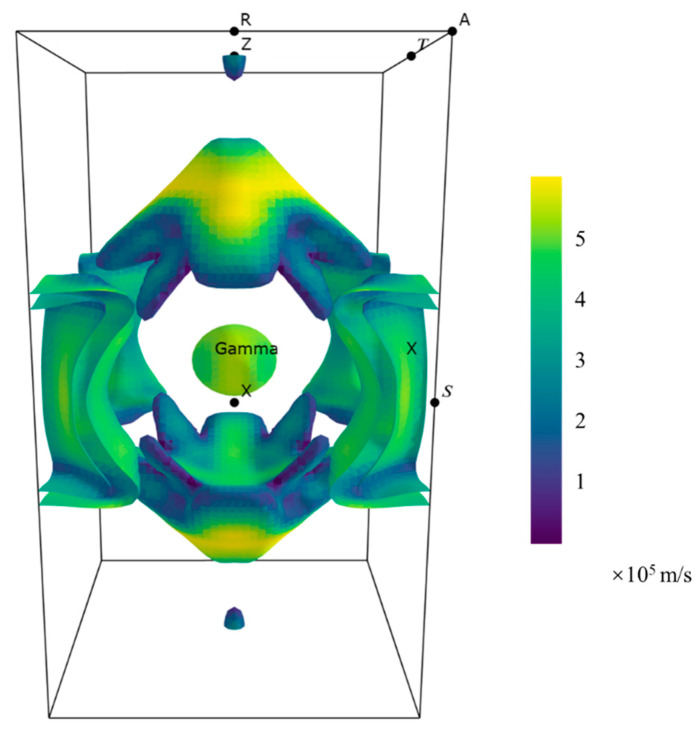
Fermi surfaces of NdB_4_ with SOC and U = 2 eV.

**Table 1 materials-16-02627-t001:** Site designations, symmetries, and atomic positions of atoms in NdB_4_ crystal.

Atom	Wyckoff Position	Symmetries	Atomic Positions
Nd	4*h*	*mm*	(x, 1/2 + x, 0)
B1	4*e*	4	(0, 0, z)
B2	8*i*	*m*	(x, y, 1/2)
B3	4*g*	*mm*	(x, 1/2 + x, 1/2)

**Table 2 materials-16-02627-t002:** Lattice parameters (Å) for various magnetic configurations of NdB_4_, calculated energies (E0), relative energies (ΔE), spin ($m_{spin}$) and orbital ($m_{orb}$) magnetic moments (in unit of μB) of different magnetic structures for NdB4. The unit for E0 and ΔE is in eV per formula unit (eV/f.u.) and in calculating ΔE the energy of 6-AFM is set to zero.

Num.	a(Å)	c(Å)	E0 (eV/f.u.)	ΔE (meV/f.u.)	$m_{spin}$ $\left(m_{orb}\right)$
1-AFM	7.195	4.092	−38.03023	20.37	±2.11(∓1.14)
2-AFM	7.196	4.084	−38.00091	49.69	±2.11(∓1.23)
3-AFM	7.196	4.083	−38.00205	48.55	±2.11(∓1.19)
4-AFM	7.194	4.092	−38.03067	19.93	±2.11(∓1.15)
5-AFM	7.203	4.086	−38.04874	1.86	±2.98(∓1.41)
**6-AFM**	**7.204**	**4.086**	−**38.05060**	**0**	**±2.99(∓1.47)**
7-AFM	7.202	4.091	−38.01553	35.07	±2.96(∓1.63)
8-AFM	7.204	4.086	−38.05016	0.44	±2.99(∓1.45)
9-AFM	7.185	4.086	−38.05016	0.44	±2.99(∓1.45)
10-AFM	7.196	4.089	−38.00589	44.71	±2.96(∓1.51)
11-NM	7.172	4.080	−36.88883	1161.77	
12-FM	7.191	4.091	−38.03820	12.40	2.98(−1.60)
13-FM	7.190	4.091	−38.03818	12.42	2.99(−1.60)
14-FM	7.189	4.091	−38.03269	17.91	3.00(−1.40)
Initial structure	7.223	4.115			
Exp. 1 (T = 30 K) [24]	7.2346	4.1101			
Exp. 2 [40]	7.220	4.102			
Exp. 3 [41]	7.219	4.1020			
Exp. 4 (T = 295 K) [42]	7.1775	4.0996			

## Data Availability

The data presented in this study are available on request from the corresponding authors. The data are not publicly available due to ongoing research in the project.

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
