# Peer review of "First-Principles Study of the Magnetic and Electronic Structure of NdB_4"

_materials, 2023, doi:10.3390/ma16072627_

Round 1
Reviewer 1 Report
The article reports on the ab-initio calculations (VASP, using PBE method for exchange-correlation and SOC included) for NdB4. The most stable spin configuration and the magnetic moments are determined.
Then, the electronic structures for the most stable spin configurations are determined by GGA+SOC and GGA+SOC+U.
Electronic structures (DOS, dispersion relation E(k), Fermi surface) are determined by GGA(SOC)+U for a couple of spin configurations.
Reviewer comments:
-The reviewer finds the Hubbard U choice not enough founded. The authors claim that Hubbard U was chosen to be 2 eV as this value gives the best fit for the calculated lattice constants. On the other side, the lattice constants they report in the paper are calculated by GGA+SOC (Table 2). Please make the U choice clearer. Also, please account for alternative methods for getting the more appropriate Hubbard U parameter for Nd (ex. A. Shankar et al, J of Semiconductors 8 082001 2012). If indeed U is 2 eV (which is unusual small) please find at least a reference in literature for such U used for Nd in similar systems.
- The GGA+U calculations description has been omitted in the section Computational details. Please add the appropriate description.
-It is not clear which approach -GGA(SOC) or GGA(SOC)+U was used for Fermi surface calculation. Please add an explanation.
-As GGA+U describes better the electronic structure of correlated system NdB4, which was the reason to perform total energy calculations for most stable spin configuration only by GGA(including SOC)approach?
The reviewer finds the present article suitable for “Materials” MDPI Journal, after dealing properly with the raised issues.
Author Response
Dear Reviewer,
Thank you very much for you for the instructive suggestions on our manuscript of “First-principles study of the magnetic and electronic structure of NdB4”. We have tried our best to revise our manuscript to comply with your comments, and the responses are itemized as follows. If there are more comments, please contact us. We will do our best to revise it again.
With best regards,
Xiaohong Shao

Reviewer 2 Report
Review report attached as a PDF file.

Author Response

(The authors gave the same response as above.)

Reviewer 3 Report
See the attached file

Author Response

(The authors gave the same response as above.)

Reviewer 4 Report
In this paper the authors investigate first principals study of NdB4 to model the magnetic and electronic structure of NdB4. I would like to request the authors to address the following concerns before I can recommend this for publication:
1. Please improve the overall quality of English. There were quite a few grammatically incorrect sentences in the paper.
2. There is no clear indication about the potential necessity and applications of this materials that would require this study. Please elaborate more about the potential applications of the material.
Author Response

(The authors gave the same response as above.)

Round 2
Reviewer 1 Report
The authors authors gave adequate answers for the issues raised by the reviewer.Also, the manuscript was modified accordingly.
As consequence, the reviewer recommend the publication of the manuscript.